# The Leukotriene Receptor Antagonist Montelukast Attenuates Neuroinflammation and Affects Cognition in Transgenic 5xFAD Mice

**DOI:** 10.3390/ijms22052782

**Published:** 2021-03-09

**Authors:** Johanna Michael, Julia Zirknitzer, Michael Stefan Unger, Rodolphe Poupardin, Tanja Rieß, Nadine Paiement, Horst Zerbe, Birgit Hutter-Paier, Herbert Reitsamer, Ludwig Aigner

**Affiliations:** 1Institute of Molecular Regenerative Medicine, Paracelsus Medical University, 5020 Salzburg, Austria; johanna.michael@pmu.ac.at (J.M.); julia.zirknitzer@alumni.pmu.ac.at (J.Z.); michael.unger@pmu.ac.at (M.S.U.); tanja_riess@gmx.de (T.R.); 2Spinal Cord Injury and Tissue Regeneration Center Salzburg (SCI-TReCS), Paracelsus Medical University, 5020 Salzburg, Austria; rodolphe.poupardin@pmu.ac.at; 3IntelgenX Corp., Saint-Laurent, QC H4S 1Y2, Canada; nadine@intelgenx.com (N.P.); horst@intelgenx.com (H.Z.); 4QPS Neuropharmacology, 8074 Grambach/Graz, Austria; birgit.hutter-paier@qps.com; 5Research Program for Experimental Ophthalmology, Department of Ophthalmology and Optometry, University Hospital of the Paracelsus Medical University, 5020 Salzburg, Austria; herbert.reitsamer@pmu.ac.at; 6Austrian Cluster of Tissue Regeneration, 1200 Vienna, Austria

**Keywords:** leukotriene receptor antagonist, cysteinyl leukotrienes, montelukast, 5xFAD, cognition, Alzheimer’s disease, microglia, RNAseq

## Abstract

Alzheimer’s disease (AD) is the most common form of dementia. In particular, neuroinflammation, mediated by microglia cells but also through CD8+ T-cells, actively contributes to disease pathology. Leukotrienes are involved in neuroinflammation and in the pathological hallmarks of AD. In consequence, leukotriene signaling—more specifically, the leukotriene receptors—has been recognized as a potential drug target to ameliorate AD pathology. Here, we analyzed the effects of the leukotriene receptor antagonist montelukast (MTK) on hippocampal gene expression in 5xFAD mice, a commonly used transgenic AD mouse model. We identified glial activation and neuroinflammation as the main pathways modulated by MTK. The treatment increased the number of Tmem119+ microglia and downregulated genes related to AD-associated microglia and to lipid droplet-accumulating microglia, suggesting that the MTK treatment targets and modulates microglia phenotypes in the disease model compared to the vehicle. MTK treatment further reduced infiltration of CD8+T-cells into the brain parenchyma. Finally, MTK treatment resulted in improved cognitive functions. In summary, we provide a proof of concept for MTK to be a potential drug candidate for AD and provide novel modes of action via modulation of microglia and CD8+ T-cells. Of note, 5xFAD females showed a more severe pathology, and in consequence, MTK treatment had a more pronounced effect in the females compared to the males. The effects on neuroinflammation, i.e., microglia and CD8+ T-cells, as well as the effects on cognitive outcome, were dose-dependent, therefore arguing for the use of higher doses of MTK in AD clinical trials compared to the approved asthma dose.

## 1. Introduction

Alzheimer’s disease (AD), the most common form of dementia, is a chronic neurodegenerative disease primarily of the aged leading to a progressive loss of cognitive abilities. Besides the deposition of amyloid plaques and the formation of neurofibrillary tangles, neuroinflammation has been recognized as a major contributor to the development of AD and, in consequence, as a target for AD therapy development [1,2,3]. Neuroinflammation is a complex molecular and cellular status of inflammatory responses in the brain and is exerted primarily by the innate and the adaptive immune system. Within the brain, the inflammatory processes are shaped by innate immunity in the form of highly dynamic microglia cells (reviewed in [4,5]) that are responsible for the release of pro- and anti-inflammatory substances, as well as for phagocytosis of cellular debris, a process which is increasingly disrupted during the course of AD [6,7]. More recently, increasing awareness arose for a putative role of the adaptive immune system, for example, T-lymphocytes that are involved in the pathogenesis of AD. T-cells invade the aged and neurodegenerated brain and thereby contribute to disease pathology ([8,9,10] and reviewed in [11]). More specifically, a population of effector memory CD8+ T-cells (T_EMRA_) present in the blood and in the cerebrospinal fluid (CSF) of AD patients negatively correlates with the cognitive scores in MCI/AD patients [12]. Infiltrating CD8+ T-cells are tightly associated with glial cell structures including microglia, forming an immune synapse, and with neuronal structures [10,12,13]. Apparently, these CD8+ T-cells regulate the expression of neuronal and synapse-related genes [14]. Undoubtedly, the specific role of T-cells in the pathogenesis of AD is far from being understood; nevertheless, the findings clearly suggest that cells of the adaptive immune system are interesting targets for future therapy development.

Leukotrienes (LTs) are lipid mediators of inflammation and have lately been suggested as a drug target in neurodegenerative diseases such as AD [15,16]. LT synthesis belongs to the complex metabolism of arachidonic acid (AA), which is cleaved from plasmamembranes by phospholipase 2. Briefly, AA is converted into 5-hydroxy-peroxy-eicosatetraenoic acid (5-HPETE) and subsequently into leukotriene A4 (LTA4), by the enzyme 5-Lox together with its activating protein FLAP [17]. LTA4 is further processed either into leukotriene B4 (LTB4) by epoxide hydrolase or into the cysteinyl leukotrienes (CysLTs) LTC4, LTD4 and LTE4 (Figure 1) [18]. CysLTs bind with different affinities to the receptors cysteinyl leukotriene receptors 1 and 2 (CysLT1R [19] and CysLT2R [20]) and G protein-coupled receptor 17 (GPR17) [21] (Figure 1A). LT signaling leads to various pro-inflammatory biological responses, such as leukocyte migration, proliferation and the production of pro-inflammatory cytokines and reactive oxygen species (ROS) (reviewed in [22]). In the context of AD, expression of 5-lipoxygenase (5-Lox), the rate-limiting enzyme in the LT pathway, is elevated in the hippocampus of the Tg2576 amyloid model and in the hippocampus of AD patients [23,24] and was found in microglia and neurons of APP-PS1 mice and AD patients [25]. Importantly, a small nucleotide polymorphism (SNP) in the gene *Alox5ap*, encoding for the 5-Lox activating protein (FLAP), is associated with an increased risk for AD [26]. Further, microglia have recently been identified as a main source of LTs in the brain of APP-PS1 mice and AD patients [25]. LTs induce neuroinflammation and blood-brain barrier (BBB) disruption, enhance neurodegeneration and inhibit neurogenesis, via signaling through LT receptors (CysLT1R, CysLT2R, GPR17), which are expressed in the brain, particularly on microglia, neurons and stem and progenitor cells [27]. Inhibition of the LT pathway through various means has illustrated preclinical proof of concept in animal models of AD. For example, genetic 5-Lox deficiency reduced plaque and Tau pathology, rescued synaptic function and improved memory in the 3 × Tg and Tg2576 mouse models of AD [23,28]. Conversely, overexpression of 5-Lox worsened the phenotype in a model of tau pathology [29]. Pharmacological inhibition of LT signaling is possible at the level of LT synthesis by blocking parts of the 5-Lox/FLAP complex or at the level of LT signaling by antagonizing the CysLT receptors. The former has demonstrated efficacy in several mouse models of AD and Tau pathology to reduce plaque and/or tau pathology and improve cognitive performance (reviewed in [16]). Furthermore, blocking of LT synthesis had an impact on neuroinflammation, as shown in a reduction in astrocytic and microglial activity [28,30,31,32,33], and a reduction in IL-1β levels [34]. The latter has shown neuroprotective effects in models of brain ischemia [35] and Parkinson’s disease (PD) [36] by inhibiting microglial reactivation, and reduced levels of reactive oxygen species in a rat model of kainic acid-induced cognitive impairment [37] and in vitro in activated human neutrophils [38]. Furthermore, inhibition at the receptor level improved cognitive performance in several mouse models [39] and reduced the levels of the pro-inflammatory cytokines TNF-a and IL-1β after intra-cerebral injection of Ab1-42 [40,41].

A straightforward approach to target LT signaling is through repurposing of montelukast (MTK), an approved LT receptor antagonist, anti-inflammatory drug and anti-asthma medicine. A 6-week oral treatment of aged rats, resembling a model for mild cognitive impairment, with MTK elevated hippocampal neurogenesis, reduced neuroinflammation, reconstituted the BBB and, most importantly, improved spatial learning and memory [27]. Furthermore, MTK has shown beneficial effects on cognition and autophagy in a mouse model for dementia with Lewy bodies [42]. MTK might indeed prevent disease and/or improve cognitive function in AD patients, which has been suggested in a retrospective analysis of the Norwegian Prescription Database (NorPD), and data from the Tromsö study [43,44]. Moreover, a recent case study with 17 dementia patients illustrated beneficial effects of MTK on cognition and agitation [45]. Interestingly, this study suggested that higher doses of MTK than the approved 10 mg/kg to treat asthma might have a higher efficacy to improve cognitive functions. Here, we aimed to illustrate the effects of prolonged MTK treatment by daily treatment of transgenic 5xFAD mice with a low or high dose of MTK. MTK or placebo was given for 13 weeks in the form of a novel mucoadhesive film with improved bioavailability and blood–brain barrier penetrance, as previously shown [46]. Using RNAseq technology and biochemical methods, we analyzed the effects of MTK on cellular and molecular levels. Furthermore, we demonstrate the efficacy of MTK to improve cognitive function in AD transgenic mice with advanced pathology using cognitive behavioral tests.

## 2. Results

### 2.1. MTK Treatment Modulates Microglia- and Neuroinflammation-Associated Gene Expression in 5xFAD Mice

#### Gene Expression Analysis

The present study aimed to illustrate the effects of prolonged MTK treatment on neuroinflammation and cognition in transgenic AD mice. Therefore, we treated transgenic 5xFAD mice with a low (3.3 mg/kg/d) or high (10 mg/kg/d) dose of MTK or placebo for 13 weeks starting at 5 months of age, where these animals already have a pronounced AD pathology (Figure 1B). With this study, we are particularly focusing on potential dose-dependent MTK effects compared to vehicle treatment in the disease model. To identify possible modes of actions of MTK in the brain, we performed whole genome transcriptomic profiling (RNA sequencing–RNAseq analysis) of hippocampal tissue from vehicle- and high-dose MTK (10 mg/kg/d)-treated transgenic 5xFAD animals. RNAseq analysis revealed 1805 significantly differentially expressed genes (DEG) (Figure 2A).

In total, 59% (1061 genes) of the DEGs were significantly downregulated and 41% (744 genes) of the DEGs were upregulated in the MTK group. The top 50 up- and downregulated genes are presented in a heatmap (Figure 2B). The top three most significantly up- or downregulated genes are depicted in Table 1. This includes *GPR17*, a gene encoding for the CysLT receptor GPR17 which is a target of MTK, as the top downregulated gene, and *Zfp46*, a zinc finger protein related to regulation of transcription, as the top upregulated gene. According to publicly available databases, zfp46 mRNA is mostly expressed in neurons and endothelial cells in the mouse brain [47] and its relative expression has been shown to be increased upon a calorie-restrictive diet in mice which impacted brain aging [48].

Functional consequences of MTK treatment.

Next, we used bioinformatics tools to obtain first insights into the functional consequences of the MTK treatment. We analyzed the DEGs for their relevance in specific fields such as AD and innate and adaptive immunity using Biomart. This annotated several of the DEGs to one or more of the following categories. (i) genes that are upregulated in AD brains [49], for example, *Fam107* and *Itpkb* (Table 2); (ii) genes annotated to innate immunity and microglia, for example, *Entpd1* [50] or mpeg1, which was shown to be increased in APPPS1 mice [51] (Table 3); and (iii) genes annotated to adaptive immunity and T-cells, for example, Laptm5, which is associated with abnormal T-cell activation and increased T-cell proliferation and identified as a new genetic risk locus for AD [52] (Table 4). Along this line, we identified *Vsir*, which encodes for a suppressor of T-cell activation expressed on microglia and elevated in AD patients [53], to be present in these categories. Gene ontology analysis identified a number of biological processes that are downregulated with MTK treatment (Appendix A) with “glial cell activation” being the top hit (Appendix A). Other significantly downregulated biological processes were “leukocyte migration”, “positive regulation of cell migration”, “ROS metabolism” and “platelet activation”. Genes annotated to these biological processes are presented in a heatmap summarized under the broad term of inflammation (Figure 2C). The second largest group of downregulated biological processes is related to regulation of synapses and vesicle transport with processes such as “regulation of synaptic vesicle cycle”, “synaptic vesicle cycle”, “vesicle-mediated transport in synapse” and “regulation of neurotransmitter levels”. Genes annotated to the pathways related to synaptic and vesicular activity are shown in a heatmap (Figure 2C). In summary, the MTK treatment, compared to the vehicle, downregulated genes associated with glial cell activation and with neuroinflammation in the 5xFAD hippocampus.

In humans, the risk of developing AD is generally higher in women than in man, with a 1.9 times higher prevalence in women in the age group 60–69 [54]. For the mouse model used in this study, sex differences in gene expression have been described, pointing towards a stronger impact of pathology in female mice [55,56]. These findings had us analyze the set of DEGs in a sex-specific manner (Figure 2B). A separation of male (blue) and female (pink) animals can be seen in both the treatment and vehicle groups, where we observed that the untreated female mice exhibit higher row Z scores, supporting the hypothesis that female mice exhibit a more pronounced pathology than males of the same age with regard to the gene expression in this heatmap (Figure 2C). Volcano plots of the sex-specific analysis show that the MTK treatment affects a very specific proportion of genes in females and a broader spectrum of genes in males (Appendix A). Genes related to AD and innate and adaptive immunity were significantly differentially expressed in both sexes (Appendix A, red gene names). Our results led us to the conclusion that (i) there are sex differences regarding gene expression in untreated 5xFAD, which is in line with the current literature, and (ii) MTK treatment has sex-specific effects on gene expression in tg 5xFAD mice.

### 2.2. MTK Treatment Boosts the Homeostatic Phenotype in Microglia

Microglia express CysLTR1 and GPR17 [27] and therefore are a cellular target of MTK. Furthermore, many of the genes modulated by the MTK treatment in the present study are related to glial cell activation and neuroinflammation. Therefore, we analyzed the transcriptome data more specifically for microglia-associated genes and hypothesized that some of the downregulated genes might indeed be expressed by microglia. We compared the dataset of MTK treatment-induced significantly downregulated genes to publicly available gene expression datasets specific for microglia subpopulations such as disease-associated microglia (DAM) [57], AD-associated microglia [58] and lipid droplet-accumulating microglia (LDAM) [59]. The latter has recently been identified as an age-related microglia phenotype with implications for AD [59]. Interestingly, we found a high overlap of the MTK downregulated genes with genes that are upregulated in AD-associated microglia (72 genes) and in LDAMs (56 genes) (Figure 3A). Furthermore, a slight overlap with genes that are upregulated in DAMs (eight genes) was observed (Appendix A). This suggests that genes that are elevated in subpopulations of AD-associated microglia and LDAMs are downregulated by the MTK treatment.

#### Histological Analysis of Microglia

Next, we analyzed the effect of the MTK treatment on microglia numbers, morphology and activation state in the hippocampus and cortex of transgenic 5xFAD mice. Iba1 (ionized calcium-binding adapter molecule 1) was used as a pan-marker to stain for microglia and macrophages [60] and to perform quantitative and morphological analysis. First, and surprisingly, we found a dose-dependent increase in the overall number of Iba1+ cells in the hippocampus of MTK-treated mice compared to vehicle-treated mice (Figure 3B,C). This information, together with the bioinformatic data (Figure 3A), triggered the use of Tmem119, a marker expressed by tissue-resident microglia [61]. Tmem119 has also been used to identify homeostatic microglia [57,62,63,64]. Therefore, we quantified the number of Tmem119− (Figure 3D) and Tmem119+ (Figure 3E) microglia in the Iba1+ population to investigate the effects of MTK on different microglia subpopulations. We found a significant and dose-dependent increase in the number of Iba1+ cells that were positive for Tmem119, but not in Iba1+ cells negative for Tmem119 (Figure 3E). This indicates that the increase in Iba1+ cells derives from an increase in the Tmem119+ subpopulation in the Iba1+ cell population, suggesting that the MTK treatment generated a more homeostatic microglia population in the 5xFAD hippocampus. Furthermore, a spatial analysis revealed that Iba1+/Tmem119+ were mainly located in plaque-distant areas, whereas Iba1+/Tmem119− cells were located directly at plaque sites (Figure 3B, arrows). We also analyzed the data regarding gender and found that female mice have significantly higher numbers of Iba1+ cells compared to male mice in all groups (Figure 3F–H), supporting differences in pathology progression between sexes in 5xFAD mice. In addition, MTK treatment resulted in a significant increase in Iba1+ cells, which derive from the Tmem119+ subpopulation in females (Figure 3F,H). Similar results were obtained in the cortex (Appendix A), where also an increase in Iba1+/Tmem119− cells was observed in the females (Appendix A).

To further characterize the activation state of microglia and macrophages in response to the MTK treatment, we analyzed microglia soma sizes as a commonly used marker for microglia activation, where a bigger soma size is indicative of aging and neurodegeneration-associated microglia. The soma sizes of Iba1+/Tmem119− and of Iba1+/Tmem119+ microglia were similar in the vehicle treatment group. However, in the high-dose MTK-treated group, specifically the Iba1+/Tmem119+ subpopulation had a smaller soma size compared to the Iba1+/Tmem119+ subpopulation in the vehicle group (Figure 4A,B), again suggesting that MTK affects the Tmem119+ population of microglia cells more specifically. The gender-specific analysis showed that specifically in the females and in the MTK group, Iba1+/Tmem119+ cells had a smaller soma size compared to Iba1+/Tmem119− cells. Furthermore, specifically in the females, MTK Iba1+/Tmem119+ cells had smaller soma sizes compared to vehicle Iba1+/Tmem119+ cells (Figure 4C). In the cortex, the effects of MTK on microglia soma size were even more pronounced, showing significantly smaller soma sizes of the Iba1+/Tmem119− population in the MTK compared to the vehicle group (Appendix A). The sex-specific analysis of the cortex data again revealed significantly smaller soma sizes in the females between MTK and vehicle Iba1+/Tmem119+ cells, and also between MTK Iba1+/Tmem119+ cells and MTK Iba1+/Tmem119− cells (Appendix A). This analysis further supports the hypothesis that MTK has pronounced effects on the Iba1+/Tmem119+ subpopulation. It also indicates that MTK might influence Iba1+ cells regardless of their Tmem119+ expression in the cortex, but not in the hippocampus. The decrease in soma size in the Tmem119 subpopulation after MTK treatment also further supports the hypothesis that MTK shifts microglia activation towards a presumably less pro-inflammatory activation state. We further hypothesized that MTK might reduce LT production as a mechanism not only to block LT signaling but also to downregulate the overall LT pathway and to lower neuroinflammation [25]. First, as we recently demonstrated, microglia are key in the production of LTs, especially under disease conditions [25]. Second, 5-Lox is the key rate-limiting enzyme in the production of LTs [17] and elevated in human AD and in AD mouse models [24,25]. Third, MTK reduces the overall level of neuroinflammation in the brain [65]. Therefore, we quantified the number of 5-Lox-positive cells and calculated the percentage of 5-Lox+ cells in the Iba1+ cell population (Figure 4D). We found no significant difference in the total number of 5-Lox+ cells (Figure 4E), but the percentage of 5-Lox+ cells in the Iba1+ microglia and macrophage population was significantly decreased by approximately ten percent in the low and high dose-treated groups compared to the vehicle-treated group (Figure 4F). Similar results were obtained in the cortex (Appendix A). This suggests that MTK treatment leads to an overall reduction in LT synthesis, at least in transgenic mice, which might contribute to a reduced level of neuroinflammation.

In AD brains, microglia are important, among other things, for the clearance of amyloid beta plaques via phagocytosis and enzymatic degradation [5,66]. To examine possible microglia-mediated or general effects of MTK treatment on progressed plaque pathology in AD, we analyzed the number as well as the area of plaques in all groups based on Thioflavin S staining (Appendix A). Plaque numbers as well as the total area covered by Thioflavin S staining did not differ between groups (Appendix A). However, female mice exhibited stronger plaque pathology than males of the same age based on plaque number and area (Appendix A). Similar results were seen in the cortex (Appendix A).

### 2.3. Montelukast Decreases the Number of CD8+ T-Cells in the Brain Parenchyma

Besides the innate immune system exemplified by microglia, there is increasing evidence that the adaptive immune system, particularly T-cells, plays an important role in modulating AD pathology. For example, T-cells are known to infiltrate the brain during aging [67] and in AD [12,14], and the majority of cells detected in the brain of AD transgenic mice were identified as CD8+ cytotoxic T-cells, outnumbering the CD4+ T-cells [8,13]. In the brain of 5xFAD mice, CD8+ T-cells have been shown to be present, and their number increases with age [9]. Once in the brain, CD8+ T-cells presumably contribute to synaptic plasticity and/or neuronal functions within the hippocampus [12,14,68]. Here, we analyzed CD8+ T-cells in the brain of 5xFAD mice and the effects of MTK treatment on such CD8+ T-cells. We discriminated between cells that infiltrated the brain parenchyma (Figure 5A, arrows) and cells that are in the brain but still associated with blood vessels using immunohistochemistry. Differentiation into vessel-associated and brain parenchyma-located T-cells was conducted based on co-staining with Collagen IV, a marker for the basement membrane of blood vessels (Figure 5B, arrows).

Quantitative analysis revealed significantly decreased numbers of CD8+ T-cells in the hippocampus (Figure 5C) after low- and high-dose MTK treatments compared to vehicle treatment. Out of the total CD8+ T-cell numbers, we analyzed the percentage of cells associated with blood vessels and cells located directly in the parenchyma. We observed that in the group treated with the high dose of MTK, significantly more CD8+ T-cells were vessel-associated compared to the other groups (Figure 5D). Conversely, the vehicle and low-dose treatment groups had significantly higher numbers of CD8+ T-cells in the brain parenchyma (Figure 5D) compared to the high-dose treatment group. Similar results were obtained in the cortex (Appendix A). Thus, MTK treatment might have reduced CD8+ T-cell infiltration into the brain parenchyma. A plausible mechanism might be the MTK-related restoration of the blood–brain barrier, as it is described in other animal models [69,70]. Indeed, one of the top upregulated genes identified by our RNA sequencing analysis was *Col4a3*, a gene for the basement membrane component collagen IV (Figure 3B). MTK might therefore have an impact on blood vessel structure and composition and restore BBB integrity, leading to reduced T-cell invasion into the brain.

### 2.4. Montelukast Treatment Dose-Dependently Improved Cognitive Functions in 5xfad Mice

Finally, our aim was to test the effects of MTK treatment on cognitive function in the transgenic 5xFAD mouse model. To assess spatial learning capacities and memory, all mice were tested in the Morris water maze (MWM) test and a cohort of seven–eight mice per group was additionally tested in the Barnes maze (BM).

#### 2.4.1. Barnes Maze

In this test, mice are supposed to learn and remember the location of a target hole on a round, open platform. Spatial learning in the BM takes advantage of the preference of mice for dark and enclosed rooms over lightened open areas. Additional white noise was played to introduce a higher motivation for the 5xFAD mice to escape the platform, which was, in general, low in the absence of white noise. Over a learning phase of four days, mice treated with high-dose MTK performed better than the vehicle group. This difference was statistically significant on day 3 between vehicle and high dose and on day 4 between both treatment groups and the vehicle group (Figure 6A). Overall, the learning effect in the BM test was small, when comparing the latency to the first target contact from day one and day four in both MTK treatment groups (Figure 6A), possibly due to a lack of motivation of the mice. Nevertheless, when analyzing the mean area under the curve of days 3–4 of the learning phase, a dose-dependent decrease was observed (*p* = 0.07), which suggests shorter latencies with increasing MTK dose (Figure 6B). Further analysis showed that mice treated with high-dose MTK had higher frequencies of target contact than both other groups. This difference was significant between the vehicle group and the high-dose MTK treatment group on day 3 (Figure 6C). On day 3 and day 4 of the experiment, the mean area under the curve of the vehicle group was significantly lower than the high-dose MTK treatment group (Figure 6D). These significant differences in the learning parameters between the vehicle and high-dose MTK treatment groups demonstrate that MTK treatment did stimulate learning behavior in transgenic 5xFAD mice.

In the memory test (Figure 6E,F), there were no significant differences between groups in the parameters of latency to target contact (Figure 6E) and frequency of target contact (Figure 6F). During the memory test, we again observed motivational problems in some individuals in all groups. However, parameters of activity, i.e., distance (Figure 6G) and speed (Figure 6H), were significantly higher in both treatment groups compared to the vehicle group.

#### 2.4.2. Morris Water Maze

As the mice tested in the BM, irrespective of their group, did not show a strong motivation to perform, all mice were tested additionally in the Morris water maze (MWM), which is another commonly used test for spatial learning and memory. In this test, the mice had to find and remember the location of a hidden platform in a pool with water. Analysis of latency times to find the hidden platform over 4 consecutive days revealed that overall, there was no difference between the groups (Figure 7A).

However, when we analyzed the dataset in a gender-specific manner, we observed significant effects in female mice (Figure 7B). Female 5xFAD mice treated with high-dose MTK showed a better task performance in learning compared to vehicle and low-dose 5xFAD mice. This difference was significant on day 3, indicating that MTK had a positive effect on learning in female 5xFAD mice. This effect is also demonstrated in the shorter distances required to find the platform at constant swimming speed in the high-dose treatment group in female mice, which are significant on day 3 (Figure 7E). This effect on distance was not found when analyzing mice of both genders together (Figure 7D). When analyzing male mice separately, no significant difference between the groups was found in the parameters of latency (Figure 7C) and distance (Figure 7F). We compared latency and distance in female and male animals within each group and noticed a difference between the sexes in the vehicle and low-dose treatment groups, but not in the high-dose treatment group (Appendix A). This shows that, in general, female 5xFAD mice are worse performers than male mice of the same age in learning behavior tasks. This difference in learning capability might be a reason why the effect is only seen in female mice. In the memory test, on day 5, animals of all groups and irrespective of sex spent comparable amounts of time in the target quadrant (Figure 7G). Nevertheless, a slight trend was observed between the vehicle and high-dose MTK treatment groups in the latency to first entry in the target quadrant (Figure 7H). MTK had no impact on the general behavior (Appendix A) or anxiety behavior (Appendix A) of transgenic mice. Taken together, the analysis of both cognitive tests shows that prolonged high-dose MTK treatment, at least in female mice, improves cognitive performance compared to vehicle-treated groups. Despite the rather small sample numbers, we nevertheless performed Pearson’s correlation analysis to define which cellular effects of MTK correlate with the cognitive improvement in the high-dose treatment group. For the hippocampus, no significant correlations were detected, but a positive correlation between cognitive improvement and microglia soma size was observed (Appendix A). However, in the cortex, we observed a significant positive correlation between cognitive improvement and CD8+ T-cell density (Appendix A). Animals with higher CD8+ T-cell numbers in the cortex performed worse in the behavior test. We observed a similar trend for animals with increased numbers of parenchymal CD8+ T-cell numbers in the cortex (*p* = 0.0777) (Appendix A).

## 3. Discussion

MTK is a leukotriene receptor antagonist that is currently under investigation to be repurposed for AD patients (NCT03402503 and NCT03991988). It has already shown efficacy in a small off-label study in human dementia patients [45] and is a promising candidate for the prevention of dementia [43,44]. In this in vivo study, we wanted to test the efficacy of MTK as a buccal film at two different doses in 5xFAD mice compared to vehicle-treated 5xFAD mice at an already progressed state of pathology (advanced plaque pathology, cognitive deficits, impaired neurogenesis and progressed neuroinflammation [71,72]). As a limitation, we want to mention that this study did not aim to include WT vehicle- or WT MTK-treated animals, and therefore one has to be cautious with the interpretation of the beneficial effects observed by MTK. MTK was systemically applied in the form of a novel pharmaceutical formulation (IntelGenx Versa Film). It was already demonstrated in a human Phase I study that this film is safe and has an improved bioavailability compared to the Singulair^®^ tablet [46]. This film was also tested in mice and there were no adverse effects of MTK on general health during a prolonged oral treatment with the MTK oral film, and the MTK presence was proven in the CSF [46]. It has also been demonstrated in rats that MTK is present in the CSF as well as in the brain [27]; however, it is present in a much lower dose compared to blood serum levels. We observed dose-dependent effects of MTK on cognition in 5xFAD mice in two cognitive tests, i.e., the Barnes maze and the Morris water maze. The long handling period prior to the test (animals were handled daily for 12 weeks before the actual start of the cognitive tests) might have led to a lower motivation to escape from the platform during the Barnes maze test.

However, the behavioral data from our study show that, compared with the placebo, high-dose MTK treatment indeed attenuated behavioral deficits in 8.5-month-old female transgenic 5xFAD mice. In this model, transgenic mice show cognitive deficits compared to their WT littermates already at an age of 4–5 months [71]. A possible explanation for the discrepancies in the effect between sex might be that the pathology is more pronounced in female transgenic mice than in male transgenic mice in this mouse model, which we could also demonstrate herein based on the higher plaque load, higher numbers of microglia in female mice and worse performance of female mice in the behavioral tests compared to male mice. These findings are in line with the current literature on sex differences in pathology in the 5xFAD mouse model, all reporting a stronger impact of pathology on females on mRNA and/or protein levels [55,56,73]. Here, an improvement in cognitive function was only observed in females in the high-dose treatment group and not in the low-dose treatment group, which did not significantly differ from the vehicle-treated group in the Morris water maze. This indicates that the effect of MTK on cognition is dose-dependent and might be even more pronounced with a higher dose of MTK. Limitations of MTK effects on cognition due to dosing limitations have been suggested by others [45]. Moreover, a higher dose might show a higher efficacy, as the high dose of 10 mg/kg/day most likely did not reach a plateau in terms of efficacy to improve cognition. However, the mice used in this study were 8.5 months old when they were tested in the cognitive tasks. The effect of MTK on cognition might be more pronounced in older animals, as the behavioral deficits of transgenic 5xFAD mice increase with age [74]. Beneficial effects of MTK have been reported in various animal models of neurodegenerative diseases, including a model of kainic acid-induced loss of memory function, an acute Huntington’s disease model of quinolinic acid and malonic acid injection-induced degeneration of striatal neurons, a beta-amyloid injection model of AD, which was accompanied by inhibition of neuroinflammation and reduced neuronal cell death, and a mouse model of dementia with Lewy bodies [37,42,65,75]. Cognitive improvements after MTK were also demonstrated in an aging rat model [27]. Therefore, our behavioral results demonstrating effects on cognition are in line with other studies using the same compound. The mode of action of MTK at the cellular and molecular levels in neurodegenerative diseases is not yet fully understood and requires further research. LT inhibition by various means has shown pleiotropic effects in the CNS, such as the attenuation of neuroinflammation (reviewed in [16]). In the periphery, LTs are mainly produced by leukocytes [76], but different subgroups of leukocytes have different capabilities of producing LTs (reviewed in [77]). In more detail, granular leukocytes are able to synthesize LTB4 and/or CysLTs, whereas B- and T-lymphocytes are not. However, lymphocytes as well as granular leukocytes express receptors for LTB4 and/or CysLTs (reviewed in [77]). LTs can also be produced, for example, by endothelial cells [78] or platelets [79] through a process of transcellular biosynthesis, where LTA4 is shuttled from leukocytes, for example, neutrophils, to cells that do not express 5-Lox and FLAP but rather downstream enzymes of the LT synthesis pathway and can therefore produce LTB4 or CysLTs with LTA4 as a substrate. Transcellular biosynthesis of LTs was also demonstrated in vitro in rat neuronal and glial cells [80]. Results from our RNA sequencing data shown in this work reveal that MTK treatment affects the immune system in the brain at the level of microglia (innate immunity) and T-cells (adaptive immunity) and MTK downregulates the activity of glial cells. Microglia are an important source of LTs in the brain [25] and several studies have shown that genetic and pharmacologic inhibition of LT production reduces glial cell activity (based on CD45 and GFAP immunoreactivity) [28,30,31,32]. Our herein shown data further demonstrate that pharmacological inhibition of LT signaling reduces glial cell activity and leads to a reduced number of Iba1+ cells that are involved in LT production. We already recently demonstrated that the number of this microglia subpopulation (Iba1+/5-Lox+) is increased in APP-PS1 mice compared to WT mice [25]. Additionally, we observed that MTK treatment increased cell numbers of a subpopulation of microglia that are Tmem119+ and are located distant to amyloid plaques. Most likely, this microglia subpopulation therefore represents a more homeostatic microglia cell type, and MTK treatment results in a shift in microglia phenotype from LT-producing microglia to more homeostatic microglia in 5xFAD mice. This shift is also reflected at the gene level in the hippocampus, where we demonstrated downregulation of genes related to AD-associated and lipid droplet-accumulating microglia. Taken together, this could mean that the additional microglia are in an altered activation state, presumably a less pro-inflammatory state, confirming the anti-inflammatory effects of MTK on immune cells of the CNS. However, we also observed that Iba1+/Tmem119+ cells that are affected by MTK are located distant from amyloid plaques, where the plaque-associated microglia were mostly Iba1+/Tmem119−. Such difference in the spatial distribution of Iba1+/Tmem119− and Iba1+/Tmem119+ subpopulations has been described by others [10]. MTK treatment specifically affected the Iba1+/Tmem119+ and plaque-distant resident microglia populations, which is one possible explanation for the fact that we did not observe any modulation of plaque pathology. In 5xFAD mice, plaques are formed already at 1–2 months of age and plaque formation rapidly progresses with age, which means that in order to modulate plaque pathology, treatment must probably start much earlier than 5 months of age.

It has already been demonstrated in other mouse models of AD that T-cells invade the amyloid-diseased brain [8,10]. Specifically, CD8+ T-cells were detected in the brain parenchyma and in close proximity to microglia cells, forming strong immune cell interactions [10]. In AD, clonally expanded CD8+ T-cells were detected in the CSF, and increased numbers of CD8+ T-cells were detected in hippocampal AD post-mortem brain sections and were negatively correlated with the cognitive scores of patients [12,14]. Here, we observed that MTK treatment decreased the numbers of CD8+ T-cells that infiltrated the brain parenchyma. These data provide a novel mode of action of MTK to modulate brain pathology. If MTK directly affects microglia and T-cells via CysLTR receptor inhibition or if the herein observed effects are indirect is not yet clear. However, results from RNA sequencing reveal that the top downregulated gene encodes for the CysLT receptor GPR17, pointing to a possible feedback loop of MTK binding on receptor gene expression. This might be one mechanism that could explain the reduced numbers of 5-Lox-expressing microglia after MTK treatment. Additional possible modes of action of MTK include an effect on blood vessel pathology [69,70,81], on neurogenesis [27] and on synaptic activity, based on the analysis of our hippocampal RNA sequencing data from MTK-treated 5xFAD mice.

This study provides a proof of concept for MTK to be a potential drug candidate for the treatment of AD and provides novel modes of action such as the modulation of microglia phenotypes and the impairment of T-cell infiltration into the diseased brain. The effects on neuroinflammation, i.e., microglia and CD8+ T-cells, as well as the effects on cognitive outcome, were dose-dependent, therefore arguing for the use of higher MTK doses in AD clinical trials compared to the approved asthma dose. For a concluding summary, see Figure 8.

## 4. Materials and Methods

### 4.1. Compounds

MTK was provided by IntelGenx corp. The substance was manufactured in an adhesive buccal film. The film contained either no MTK as vehicle control, 1 mg/cm^2^ MTK or 3 mg/cm^2^ MTK. Films were stored at RT under dry and air-free conditions. Every day before administration, the film was cut to 7mm^2^ round, which translates to final MTK concentrations of 3.3 (= low dose) and 10 mg/kg/d (= high dose). Animals received either vehicle treatment (= film without drug) or low or high dose of MTK.

### 4.2. Animals

For this study, 45 transgenic 5xFAD mice [71], which have three mutations in the gene for APP695 (PP K670N/M671L (Swedish), I716V (Florida), V717I (London)) and two mutations in the gene for PS1 (M146L, L286V) under the expression of the Thy1 promotor, were used (available at QPS Austria). Mice were housed in groups under standard conditions at the animal facility at QPS Austria with a constant 12-h light/dark cycle at 24 °C room temperature, relative humidity of 40–70% and standard rodent chow (altrumin) and water ad libitum. The experiment was approved by local ethical committees (Amt der steiermärkischen Landesregierung, ABT13–14688/2018-4, 11.02.2018) and was conducted at QPS. In this study, 5-month-old animals were treated with MTK for 13 weeks. Daily, MTK was administered orally in the form of a mucoadhesive film, which was placed on the mice’s buccal mucosa. Animals received either vehicle treatment (= film without drug) or 3.3 (referred to as low dose) or 10 mg/kg/d MTK (referred to as high dose). Animals were tested in several behavioral tests in the last three weeks of the experiment (Figure 1A).

### 4.3. Behavioral Tests

To monitor general behavior, the open field (OF; 48 × 48 cm; TSE-System^®^, Bad Homburg, Germany) was used. Infrared photo beams were placed at a 1.4 cm distance around the box and 4 cm higher to detect rearing. Test sessions lasted for 20 min and were performed under standard room light conditions. Analysis was conducted for the whole 20 min as well as for 5 min intervals. To measure anxiety, mice were tested in the elevated plus maze (EPM). The maze was elevated 50 cm above ground and the test took place under red light conditions. The behavior was recorded for five minutes and parameters were recorded with Noldus Ethovision XT. Spatial learning capacities of all animals were tested in the Morris water maze (MWM) and Barnes maze (BM). The maze for BM was elevated 105 cm above ground and consisted of a circular platform (92 cm diameter) with 20 holes around the perimeter. During the test, animals received reinforcement (in the form of light, a fan and 80 db white noise) to escape from the open platform to a small dark enclosed target box located under the platform. Mice could access the target box through the target hole. From the center of the maze, all holes looked identical and the target box could not be visually discriminated. Visual 3D cues were placed around the maze. For quantification, a computerized video tracking system (Noldus Observer, Wageningen, the Netherlands) was used. The MWM was performed using the following pattern: Four trials were performed on four consecutive days. In all trials, the platform was located in the northeast (NE) quadrant of a pool and mice started from predefined positions. A single trial lasted for 60 s. At 24 h after the last trial on day 4, mice were tested in the probe trial (PT), during which the platform was removed. For the quantification of escape latency (the time (s) to find the hidden platform), of pathway (the length of the trajectory (m) to reach the target), of target zone crossings and of the abidance in the target quadrant in the PT, a computerized video tracking system (Biobserve, Viewer III, http://www.biobserve.com/behavioralresearch/ (accessed on 10 January 2021)) was used.

### 4.4. Perfusion and Tissue Sectioning

All animals were euthanized on the day of the last behavioral test by Pentobarbital injection (600 mg/kg). Serum, CSF and brains were collected from all animals. After deep anesthesia, blood was taken by cardiac puncture, and mice were transcardially perfused with 0.9% saline and the brains were taken out and hemisected. Right hemibrains were fixed by immersion in freshly prepared 4% paraformaldehyde/PB (pH = 7.4) for 48 h at 4 °C. Thereafter, right hemibrains were transferred to 30% sucrose in PBS solution for 24–72 h at 4 °C to ensure cryoprotection. Right hemibrains were transferred a second time to fresh 30% sterile filtered sucrose in PBS solution and stored at 4 °C until cut into 40 µm sections using a sliding microtome (Leica). The left hemisphere was dissected into the hippocampus, cortex and rest, weighed and snap frozen on dry ice in distinct tubes and stored at −80 °C.

### 4.5. Fluorescence Immunohistochemistry (IHC)

Fluorescence immunohistochemistry of mouse tissue was performed on free-floating sections as previously described [82,83]. Antigen retrieval was performed depending on the used primary antibody by steaming the sections for 15–20 min in citrate buffer (pH = 6.0, Sigma St. Louis, MO, USA). The following primary antibodies (Appendix A) were used overnight: goat anti-Iba1 (1:500, Abcam, Cambridge, MA, USA), rabbit anti-Tmem119 (1:100, Abcam, Cambridge, MA, USA), rabbit anti-5-Lox (1:100, Abcam, Cambridge, MA, USA), rat anti-CD8 (1:100, eBioscience 140195, San Diego, CA, USA), goat anti-ColIV (1:100, Millipore, Darmstadt, Germany). Sections were extensively washed in PBS and incubated for 3 h at RT in secondary antibodies all diluted at 1:1000. The following secondary antibodies were used: donkey anti-goat Alexa Fluor 647, donkey anti-guinea pig Alexa Flour 647 (Jackson Immuno Research, Cambridge, UK), donkey anti-rabbit Alexa Fluor 488, donkey anti-rabbit Alexa Fluor 568 (all Invitrogen/Life Technologies Waltham, MA, USA). Nucleus counterstaining was performed with 4′,6′-diamidino-2-phenylindole dihydrochloride hydrate (DAPI 1 mg/mL, 1:2000, Sigma St. Louis, MO, USA). Stainings of amyloid beta plaques were performed with Thioflavin S (1 mg/mL, 1:650, Sigma St. Louis, MO, USA). Tissue sections were additionally treated with 0.2% Sudan Black (Sigma St. Louis, MO, USA) in 70% ethanol for 2 min to reduce the autofluorescence in tissue from old animals [84,85,86]. After this treatment, the sections were extensively washed in PBS and mounted onto microscope glass slides (Superfrost Plus, Thermo Scientific, Waltham, MA, USA). Brain sections were cover-slipped semi-dry in ProLong Gold Antifade Mountant (Life technologies, Waltham, MA, USA).

#### 4.5.1. Quantification of Iba1^+^/5-Lox^+^ and Iba1^+^/5-Lox^−^ Cell Numbers

For quantification of 5-Lox+ cell numbers and percentage of 5-Lox+ cells in the Iba1+ cell population, we analyzed 3 confocal z-stack images at 20 × magnification of different hippocampi and cortices per animal and counted the number of Iba1^+^/5-Lox^+^ and Iba1^+^/5-Lox^−^ cells using Fiji (ImageJ 1.52p; https://imagej.net/Fiji (accessed on 10 January 2021)). The mean of 3 images per animal (*n* = 6 animals/group) was calculated. Only cells with a cell nucleus and clearly visible cell soma were included in the analysis. Percentages of Iba1^+^/5-Lox^+^ and Iba1^+^/5-Lox^−^ cell numbers in total Iba1^+^ cell counts were calculated in Excel.

#### 4.5.2. Soma Size Analysis

For soma size analysis of Iba1+/Tmem119− and Iba1+/Tmem119+ cells, we analyzed 3 confocal z-stack images at 40 × magnification of different hippocampi and cortices per animal and measured 5 cells per population per image using Fiji (ImageJ 1.5p; https://imagej.net/Fiji (accessed on 10 January 2021)). Only cells with a cell nucleus and clearly visible soma were included in the analysis. We analyzed six animals (3 males and 3 females) per group and calculated the mean of 90 cells per group. In the sex-specific analysis, we calculated the mean of 45 animals per group.

#### 4.5.3. Quantification of CD8+ T-Cells

Three whole sagittal sections with a clearly visible and complete hippocampus and cortex were analyzed per animal (*n* = 6 animals/group). Areas of the hippocampus and cortex were measured using Fiji (ImageJ 1.52p; https://imagej.net/Fiji (accessed on 10 January 2021)). For quantification of CD8+ T-cell numbers and percentage of parenchymal and vessel-associated CD8+ T-cells, we counted the number of CD8+ T-cells in the parenchyma and the number of CD8+ T-cells co-localized with ColIV staining. Percentages of parenchymal and vessel-associated CD8+ T-cells were calculated in Excel.

### 4.6. Confocal Microscopy and Image Processing

For imaging, confocal laser scanning microscopes LSM700 and LSM710 (Zeiss, Oberkochen, Germany) were used and gratefully provided by the microscopy core facility of SCI-TReCS (Spinal Cord Injury and Tissue Regeneration Center Salzburg). Images were taken as confocal z-stacks using 20 ×, 40 × and 63 × oil magnifications with 0.5, 0.6 or 1.0 zoom and combined to merged maximum intensity projections with the ZEN 2011 SP3 or SP7 (black edition) software (Zeiss, Oberkochen, Germany). For qualitative analysis, 2–3 animals per group were immunohistologically stained and analyzed. For quantitative analysis, 6 animals per group were stained and analyzed. Editing and processing of all images were performed using ZEN 2012 (blue edition) software (version 1.1.2.0 Zeiss, Oberkochen, Germany) and Microsoft PowerPoint. Three-dimensional rendering was performed using Imaris Software (version 9.1.2, Bitplane, https://imaris.oxinst.com/ (accessed on 10 January 2021)). Images of CD8+ T-cell stainings were taken with a virtual slide microscope, VS120, with the Olympus VS-ASW.L100 software (both from Olympus, https://www.olympus.de/ (accessed on 10 January 2021), Hamburg, Germany). Images were taken of whole sagittal sections as confocal z-stacks using 20× magnification.

### 4.7. RNA Isolation and RNA Sequencing Analysis

To analyze possible changes in mRNA levels of MTK-treated mice, the total RNA was extracted from the mouse hippocampus of vehicle- (*n* = 10) and high-dose MTK-treated (*n* = 10) mice as described before [25]. In brief, tissue was homogenized in 1 mL Trizol (TRI^®^Reagent; Sigma St. Louis, MO, USA). For phase separation, 150 µL of 1-bromo-3-chloropropane (Sigma) was added, vortexed and centrifuged (15 min at 12,000× *g* at 4 °C). After transferring the aqueous phase into a new tube, 500 µL 2-Propanol p.A. (Millipore) was added and vortexed. To obtain RNA, samples were centrifuged (10 min at 12,000× *g* at 4 °C). The pellet was washed with 1 mL 75% ethanol, dried and re-suspended in 30 µL RNase-free water (pre-warmed to 55 °C). cDNA was synthesized using the iScript Reverse Transcription Supermix (Bio-Rad, Hercules, CA, USA). Samples were stored at −80 °C. Whole transcriptome analysis of total RNA was performed by Qiagen Genomic Services. Analysis was performed in accordance with the company protocols outlined as follows: The library preparation was conducted using TruSeq^®^ (Illumina Inc. San Diego, CA, USA) Stranded mRNA Sample preparation kit (Illumina inc, San Diego, CA, USA). The starting material (500 ng) of total RNA was mRNA-enriched using the oligodT bead system. The isolated mRNA was subsequently fragmented using enzymatic fragmentation. Then, first-strand synthesis and second-strand synthesis were performed and the double-stranded cDNA was purified (AMPure XP, Beckman Coulter). The cDNA was end repaired and 3′ adenylated, Illumina sequencing adaptors were ligated onto the fragment ends and the library was purified (AMPure XP). The mRNA stranded libraries were pre-amplified with PCR and purified (AMPure XP). The libraries’ size distribution was validated and quality-inspected on a Bioanalyzer 2100 or BioAnalyzer 4200 tapeStation (Agilent Technologies, Waldbronn, Germany). High-quality libraries were pooled based in equimolar concentrations based on the Bioanalyzer Smear Analysis tool (Agilent Technologies). The library pool(s) was (were) quantified using qPCR and an optimal concentration of the library pool was used to generate the clusters on the surface of a flowcell before sequencing on a NextSeq500) instrument (50 cycles) according to the manufacturer’s instructions (Illumina Inc. San Diego, CA, USA). Tables of differentially expressed genes were generated with Microsoft Excel software (https://www.microsoft.com/de-de/ (accessed on 10 January 2021)).

### 4.8. RNA Sequencing Analysis and Bioinformatics

Whole transcriptome analysis was performed by Qiagen Genomic Services from total RNA. Analysis was performed in accordance with the company protocols outlined as follows: The starting material (100 ng) of total RNA was mRNA-enriched using the oligodT bead system. The isolated mRNA was subsequently fragmented using enzymatic fragmentation. Then, first-strand synthesis and second-strand synthesis were performed and the double-stranded cDNA was purified (AMPure XP, Beckman Coulter, Krefeld, Germany). The cDNA was end repaired and 3 s adenylated, Illumina sequencing adaptors were ligated onto the fragment ends and the library was purified (AMPure XP). The mRNA stranded libraries were pre-amplified with PCR and purified (AMPure XP). The library size distribution was validated and quality-inspected on a Bioanalyzer 2100 or BioAnalyzer 4200 tapeStation (Agilent Technologies). High-quality libraries were pooled based in equimolar concentrations based on the Bioanalyzer Smear Analysis tool (Agilent Technologies). The library pool(s) was (were) quantified using qPCR and an optimal concentration of the library pool was used to generate the clusters on the surface of a flowcell before sequencing on a NextSeq500) instrument (75 cycles) according to the manufacturer’s instructions (Illumina Inc. San Diego, CA, USA). Quality control of raw sequencing data was conducted using FastQC tool [85]. Reads were then mapped to the genome (*Mus musculus* genome GRCm38) using bowtie2 (version 2.2.2, [86]). Reads that overlap with genes were then counted using the HTSEQ tool (version 0.11.2, [87], -m intersection-nonempty -s no -i gene_id -t exon). Expression values of protein coding genes were first normalized and differential expression analysis between the different groups was conducted using Deseq2 [88]. Genes were considered significantly differentially transcribed with an adjusted *p*-value < 0.05 (Benjamini and Hochberg multiple testing correction). Genes were annotated using the biomaRt package [89]. Gene Ontology (GO) term and Kegg enrichment analysis were conducted using the package ClusterProfiler [90]. GO or Kegg pathways were considered significantly enriched if *p*-value < 0.05. Raw data of differentially expressed genes in total and for females and males separately can be found in the Appendix A.

### 4.9. Funrich

The software “Funrich” [91] was used for the creation of the Venn Diagram. We compared our dataset with the following datasets of specific microglia populations: LDAM [59], DAM [57], AD-associated microglia [58].

### 4.10. Statistics

For statistical analysis, the Prism 5–9 software (GraphPad https://www.graphpad.com/scientific-software/prism/ (accessed on 10 January 2021)) was used. The data were tested for normal distribution with the Kolmogorov–Smirnov test. For comparison of two groups, an unpaired *t*-test was performed. For comparison of more than two groups, one-way analysis of variance (ANOVA) was used with Tukey’s or Bonferroni’s multiple comparison test as a post hoc test. Correlation analysis was conducted using Pearson’s correlation test in GraphPad Prism. The data were depicted as mean and standard error of the mean (SEM) or standard deviation (SD) with a 95% confidence interval as indicated in the respective figure legends. *P*-values of *p* < 0.0001 and *p* < 0.001 were considered extremely significant (**** or ***), *p* < 0.01 very significant (**) and *p* < 0.05 significant (*).

## Figures and Tables

**Figure 1 ijms-22-02782-f001:**
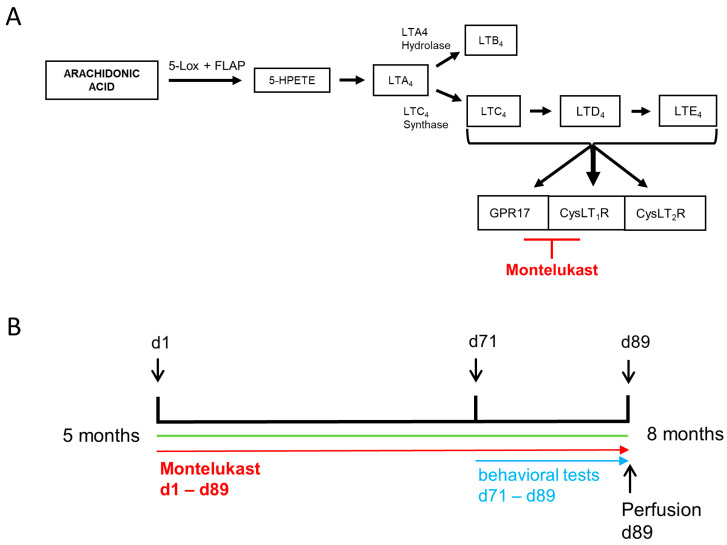
(**A**) The leukotriene signaling pathway. Leukotrienes arise from arachidonic acid (AA), which is converted to LTA4 by the enzyme 5-Lipoxygenase (5-Lox) and its activating protein (FLAP). LTA4 is either metabolized to LTB4 by LTA4 hydrolase or to LTC4 by LTC4 synthase. From LTC4, the other cysteinyl leukotrienes, i.e., LTD4 and LTE4, arise. Cysteinyl leukotrienes bind to the receptors CysLTR1, CysLTR2 and GPR17. Leukotriene signaling can be inhibited by targeting the receptors for cysteinyl leukotrienes with the leukotriene receptor antagonist montelukast. (**B**) Experimental setup. Five-month-old 5xFAD transgenic mice were treated daily with vehicle or montelukast in two different doses for 89 days. Behavioral tests were performed between day 72 and day 89. On the last day of the behavioral tests, the mice were perfused, and tissue samples were collected.

**Figure 2 ijms-22-02782-f002:**
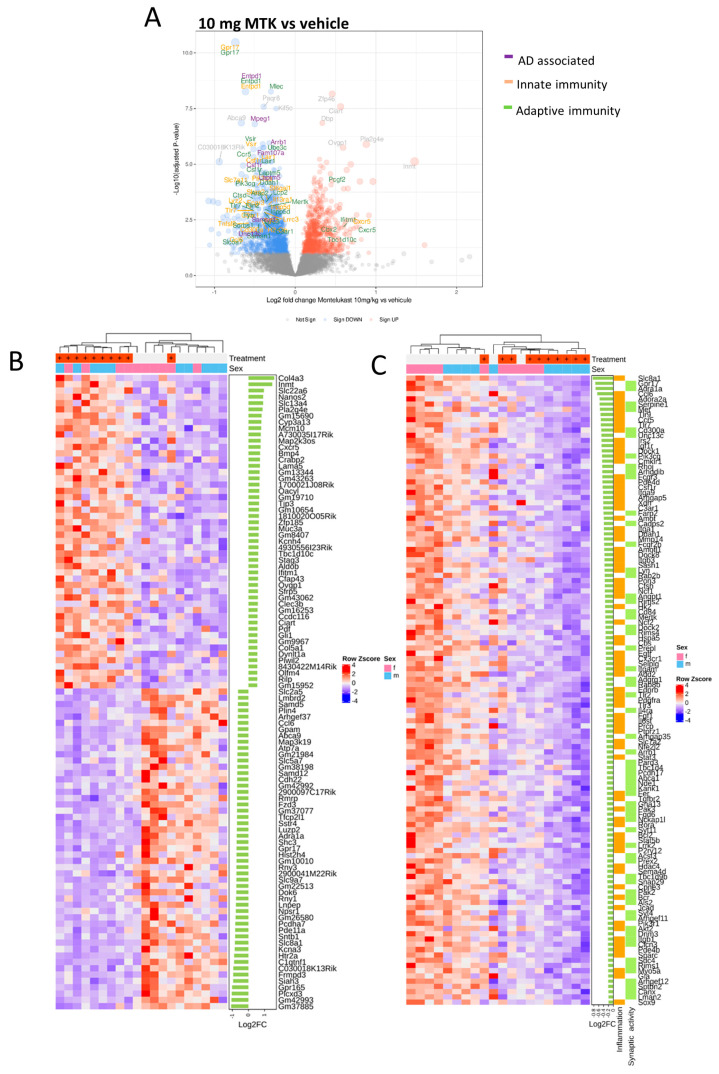
MTK alters the gene expression profile in 5xFAD mice. Gene expression data from hippocampal tissue (*n* = 10/group). Only tissues from the vehicle and high-dose (10 mg/kg/d) groups were used in this analysis. (**A**) Volcano plot showing differentially expressed genes (DEGs) in high dose- versus vehicle-treated groups. Significantly upregulated DEGs are depicted in red and significantly downregulated DEGs are depicted in blue. Gene names associated with the terms innate immunity (yellow), adaptive immunity (green) and Alzheimer’s disease (AD) (purple) are highlighted. (**B**) Heatmap showing the top 50 significantly up- and downregulated genes. (**C**) Genes from GO annotations that fit into the broad terms inflammation and synaptic activity are the top downregulated biological processes. Here, the respective genes are shown in a heatmap.

**Figure 3 ijms-22-02782-f003:**
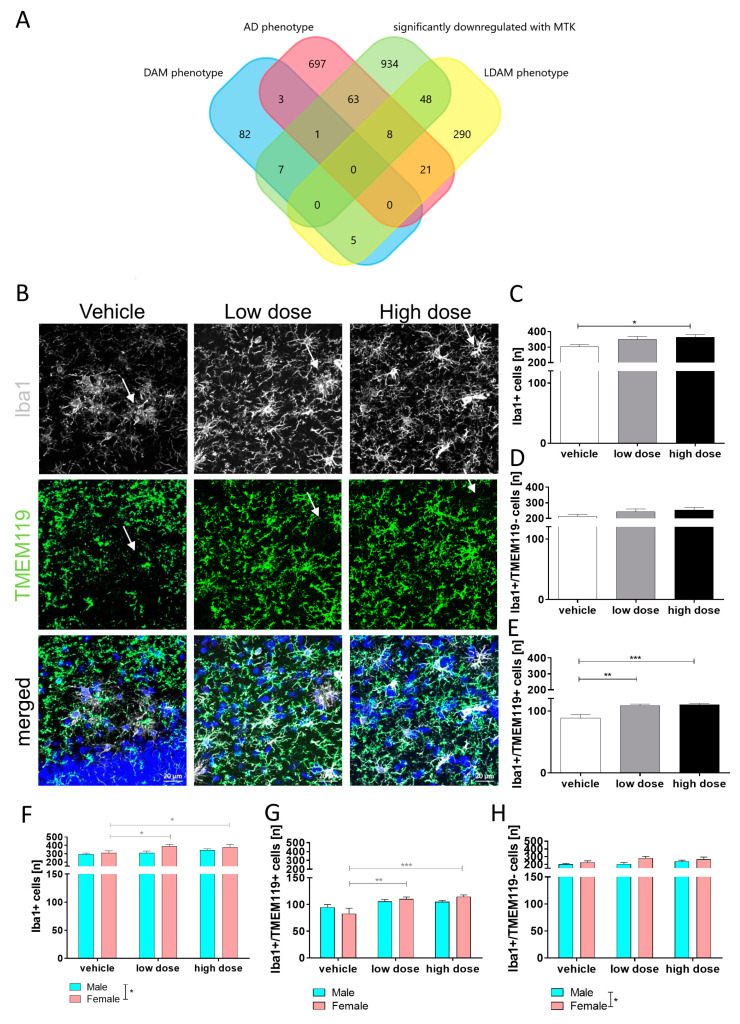
Bioinformatic and immunohistochemical analysis of microglia with and without MTK treatment in the hippocampus. (**A**) Overlap between genes downregulated with MTK treatment (green) and genes upregulated in lipid droplet-accumulating microglia (red), disease-associated microglia (blue) and AD microglia (yellow). (**B**) Representative images of microglia positive for Iba1 and Tmem119; scale bar: 20 µm. (**C**) We observed an increase in numbers of Iba1+ microglia with MTK, which was significant between vehicle and high dose. (**D**) No significant difference in numbers of Iba1+/Tmem119− cells between groups. (**E**) We observed a significant increase in numbers of Iba1+/Tmem119+ microglia with MTK. (**F**) The increase in Iba1+ cells comes from female mice, which have significantly more microglia than male mice of the same age. (**G**) Numbers of cells separated by sex. Female mice have significantly more Iba1+/Tmem119− cells than male mice. (**H**) The increase in the subpopulation comes from females, which have less of these cells in the vehicle group but increase to more of these cells in both MTK treatment groups. Data are shown as mean +/− SEM. One-way ANOVA was performed with Tukey’s post hoc test. *P*-values of *p* < 0.0001 and *p* < 0.001 were considered extremely significant (***), *p* < 0.01 very significant (**) and *p* < 0.05 significant (*).

**Figure 4 ijms-22-02782-f004:**
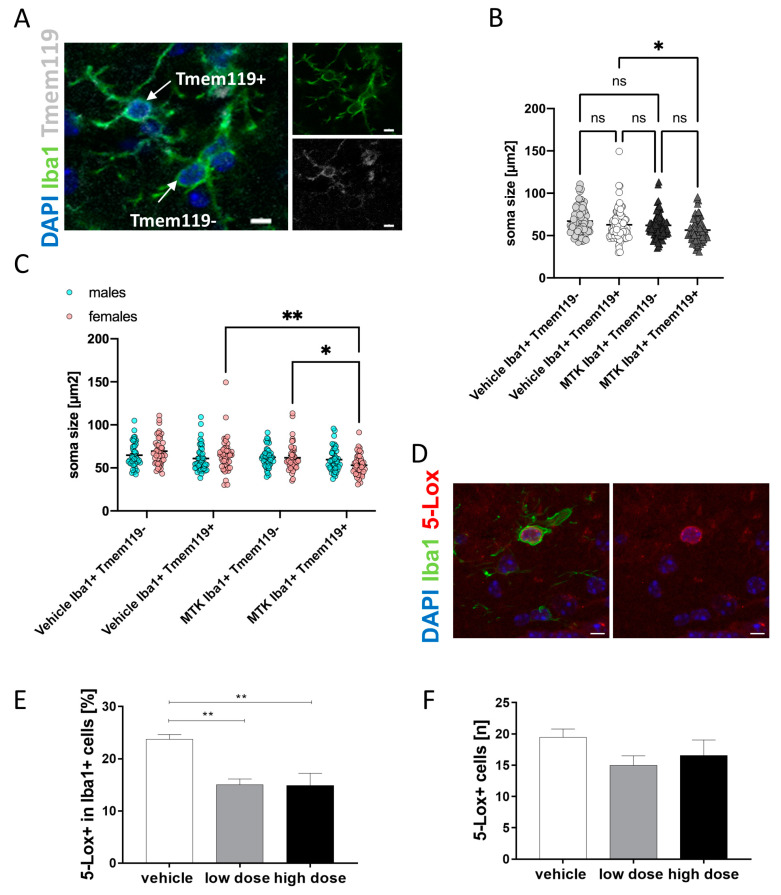
Analysis of microglia activation. (**A**) Representative image of Iba1+ microglia positive and negative for Tmem119; scale bar: 5 µm. (**B**) MTK reduces soma sizes in the Iba1+/Tmem119+ subpopulation in the hippocampus. (**C**) Sex-specific soma size analysis: effects of MTK on soma size are seen only in females in the hippocampus. (**D**) Representative image of Iba1+ microglia positive for 5-Lox; scale bar: 5 µm. (**E**) Quantification of 5-Lox-positive cells. (**F**) Percentage of 5-Lox+ cells in the Iba1+ microglia population. There was no significant difference between male and female mice in this analysis. Data are shown as mean +/− SEM. One-way ANOVA was performed with Tukey’s post hoc test. *P*-values of *p* < 0.01 were considered very significant (**) and *p* < 0.05 significant (*).

**Figure 5 ijms-22-02782-f005:**
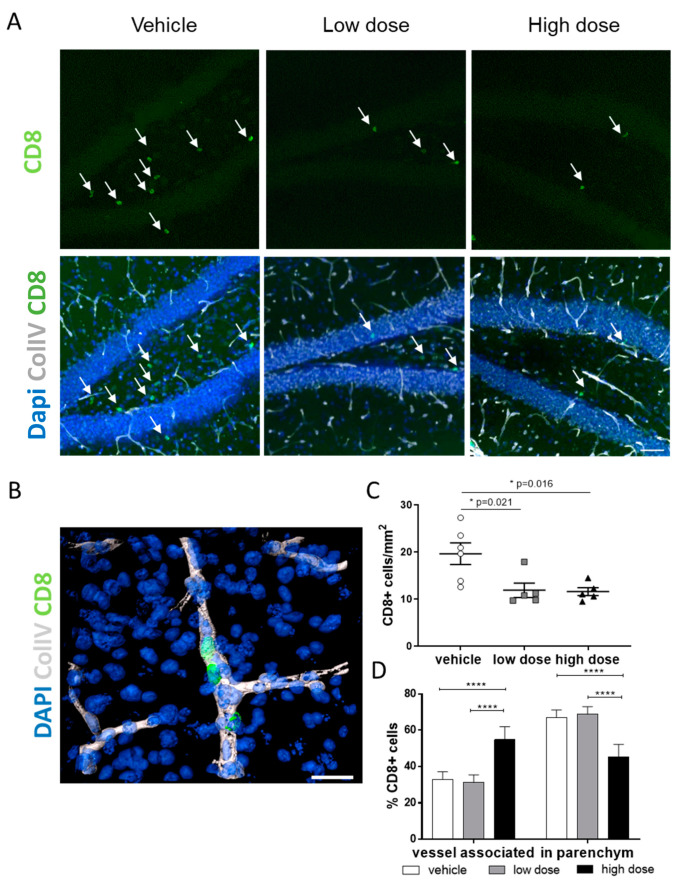
Immunohistochemical analysis of CD8+ T-cells in the hippocampus. (**A**) Representative images of parenchymal CD8+ T-cells and ColIV in the hippocampus. Scale bar: 10 µm (**B**) Representative image of vessel-associated CD8+ T-cells. Scale bar: 5 µm (**C**) Number of CD8+ T-cells per mm2 with and without MTK. (**D**) Distribution of CD8+ T-cells between vessel and parenchyma among all groups. Data are shown as mean +/− SEM. One-way ANOVA was performed with Tukey´s post hoc test. *P*-values of *p* < 0.0001 and *p* < 0.001 were considered extremely significant (****) and *p* < 0.05 significant (*). Images were created with Zen blue (**A**) and Imaris (**B**).

**Figure 6 ijms-22-02782-f006:**
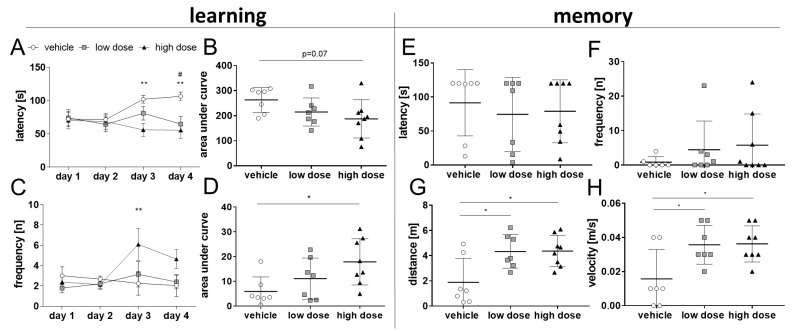
Learning behavior in the Barnes maze (BM). Mice were tested in the BM (*n* = 7–8/group). (**A**) During the learning phase of the test, mice treated with vehicle performed worse than mice treated with the high dose of MTK. Mice treated with the high dose performed significantly better than the vehicle group on days 3 and 4. (**B**) Mean area under curve (AUC) of latency on days 3–4 of the BM shows a trend to a dose-dependent decrease (*p* = 0.07). (**C**) Mice from the high-dose treatment group had higher frequencies of target contact, which was significant on day 3 between high dose- and vehicle-treated groups. (**D**) AUC of frequency on days 3–4 of the BM shows a dose-dependent increase, which is significant between vehicle and high-dose treatment groups. (**E**,**F**) During the memory test, no significant difference between groups was detected in the parameters of latency and frequency. However, mice had huge motivational problems regardless of their group. (**G**,**H**) Mice treated with MTK did show significant longer distances and higher velocities compared to vehicle-treated animals. Data are shown as mean +/− SEM. Two-way ANOVA (**A**,**C**) or one-way ANOVA (**B**,**D**,**E**–**H**) was performed with Tukey´s post hoc test. *P*-values *p* < 0.01 were considered very significant (**) and *p* < 0.05 significant (*).

**Figure 7 ijms-22-02782-f007:**
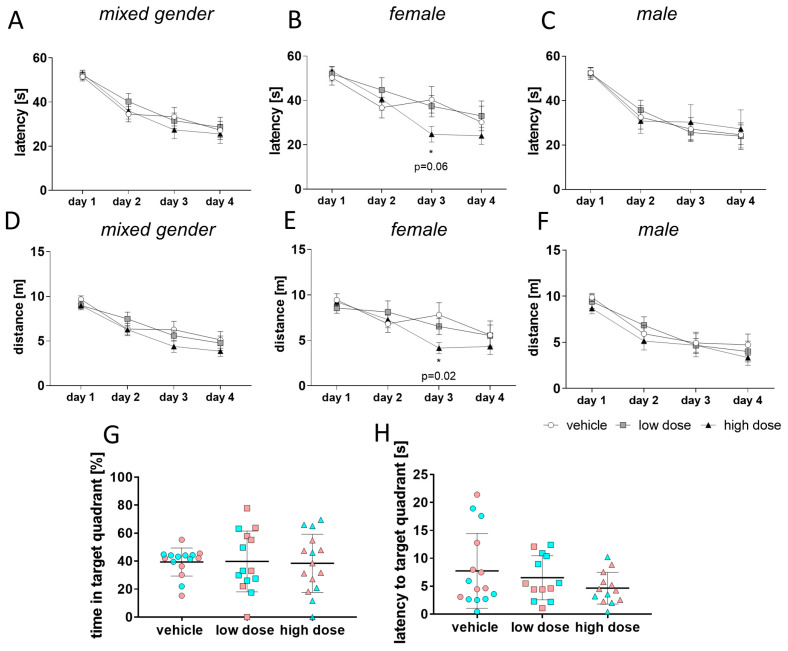
Learning and memory in the Morris water maze (MWM). Mice were tested in the MWM (*n* = 14–15/group). (**A**–**C**) Latency to target contact in the learning phase did not reveal a significant difference between groups (**A**) when tested regardless of gender. When female mice were analyzed separately (*n* = 7–8/group), a trend (*p* = 0.06) to better learning was detected on day 3 between vehicle and high-dose treatment groups (**B**). Separate analysis of male mice revealed no significant difference between groups amongst males (**C**). (**D**–**F**) Analysis of distance revealed no significant difference between groups (**D**) when tested regardless of gender. When female mice were analyzed separately, they showed shorter distances, which were significant on day 3 (**E**). Separate analysis of male mice revealed no significant difference between groups amongst males (**F**). Memory tests on day 5 revealed no significant difference between groups regardless of gender (pink = female, blue = male) (**G**,**H**). Data are shown as mean +/− SEM. Two-way ANOVA (**A**–**F**) or one-way ANOVA (**G**,**H**) was performed with Tukey’s post hoc test. *p*-values of *p* < 0.05 were considered significant (*) and *p* < 0.07 were considered as trend.

**Figure 8 ijms-22-02782-f008:**
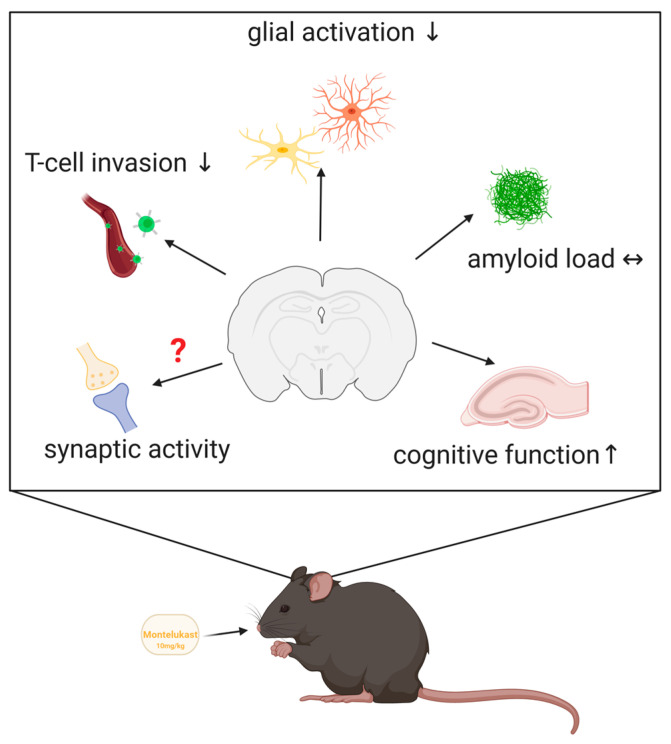
MTK has pleiotropic effects on the diseased brain. MTK treatment with 10 mg/kg/d has pleiotropic effects in the brain, i.e., reduced microglia activation and CD8+ T-cell invasion and improvement in cognition. Furthermore, MTK affects gene expression of genes related to synaptic activity.

**Table 1 ijms-22-02782-t001:** Top 3 most significantly down- and upregulated genes.

Gene	Description	Fold Change (log2)	Adjusted *p*-Value < 0.05
*Gpr17*	G protein-coupled receptor 17	−7.4 × 10^−1^	3.4 × 10^−11^
*Entpd1*	ectonucleoside triphosphate diphosphohydrolase 1	−6.2 × 10^−1^	5.5 × 10^−9^
*Mlec*	malectin	−3 × 10^−1^	5.5 × 10^−9^
*Zfp46*	zinc finger protein 46	4.6 × 10^−1^	6.9 × 10^−9^
*Ciart*	circadian associated repressor of transcription	5.6 × 10^−1^	2.6 × 10^−8^
*Dbp*	D site albumin promoter binding protein	3.4 × 10^−1^	1.4 × 10^−7^

**Table 2 ijms-22-02782-t002:** Significantly differentially expressed genes related to AD.

Gene	Description	Fold Change (log2)	Adjusted *p*-Value < 0.05
*Entpd1*	ectonucleoside triphosphate diphosphohydrolase 1	−0.62	5.47 × 10^−9^
*Mpeg1*	macrophage expressed gene 1	−0.50	1.56 × 10^−7^
*Arrb1*	arrestin, beta 1	−0.28	1.79 × 10^−6^
*Fam107a*	family with sequence similarity 107, member A	−0.41	4.90 × 10^−6^
*Csf1r*	colony stimulating factor 1 receptor	−0.44	1.82 × 10^−5^
*Laptm5*	lysosomal-associated protein transmembrane 5	−0.39	2.32 × 10^−5^
*Samsn1*	SAM domain, SH3 domain and nuclear localization signals, 1	−0.46	5.13 × 10^−3^
*Unc13c*	unc−13 homolog C	−0.49	8.11 × 10^−3^

**Table 3 ijms-22-02782-t003:** Significantly differentially expressed genes related to microglia and innate immunity.

Gene	Description	Fold Change (log2)	Adjusted *p*-Value < 0.05
*Gpr17*	G protein-coupled receptor 17	−0.74	3.41 × 10^−11^
*Entpd1*	ectonucleoside triphosphate diphosphohydrolase 1	−0.62	5.47 × 10^−9^
*Vsir*	V-set immunoregulatory receptor	−0.52	2.14 × 10^−6^
*Lair1*	leukocyte-associated Ig-like receptor 1	−0.45	7.76 × 10^−6^
*Csf1r*	colony stimulating factor 1 receptor	−0.44	1.82 × 10^−5^
*Slc7a11*	solute carrier family 7 (cationic amino acid transporter, y + system), member 11	−0.56	8.44 × 10^−5^
*Pik3cg*	phosphatidylinositol−4,5-bisphosphate 3-kinase catalytic subunit gamma	−0.47	9.25 × 10^−5^
*Skap2*	src family associated phosphoprotein 2	−0.35	3.80 × 10^−4^
*Lyz2*	lysozyme 2	−0.55	9.26 × 10^−4^
*St6gal1*	beta galactoside alpha 2,6 sialyltransferase 1	−0.35	1.65 × 10^−4^
*Il13ra1*	interleukin 13 receptor, alpha 1	−0.33	6.58 × 10^−4^
*Fcgr3*	Fc receptor, IgG, low affinity III	−0.45	1.02 × 10^−3^
*Tlr7*	toll-like receptor 7	−0.51	1.19 × 10^−3^
*Inpp5d*	inositol polyphosphate-5-phosphatase D	−0.37	1.04 × 10^−3^
*Fyb*	FYN binding protein	−0.47	1.63 × 10^−3^
*Lrrc3*	leucine rich repeat containing 3	−0.38	1.43 × 10^−3^
*C3ar1*	complement component 3a receptor 1	−0.41	2.80 × 10^−3^
*Cd300a*	CD300A molecule	−0.50	4.39 × 10^−3^
*Ccl6*	chemokine (C-C motif) ligand 6	−0.66	1.49 × 10^−2^
*Tnfsf8*	tumor necrosis factor (ligand) superfamily, member 8	−0.58	4.39 × 10^−3^
*Itgb3*	integrin beta 3	−0.36	1.46 × 10^−2^
*H2-Ob*	histocompatibility 2, O region beta locus	−0.42	3.38 × 10^−3^
*Cxcr5*	chemokine (C-X-C motif) receptor 5	0.75	6.55 × 10^−3^

**Table 4 ijms-22-02782-t004:** Significantly differentially expressed genes related to T-cells and adaptive immunity.

Gene	Description	Fold Change (log2)	Adjusted *p*-Value < 0.05
*Gpr17*	G protein-coupled receptor 17	−0.74	3.4 × 10^−11^
*Entpd1*	ectonucleoside triphosphate diphosphohydrolase 1	−0.62	5.47 × 10^−9^
*Mlec*	malectin	−0.30	5.47 × 10^−9^
*Vsir*	V-set immunoregulatory receptor	−0.52	2.14 × 10^−6^
*Ube3c*	ubiquitin protein ligase E3C	−0.29	3.04 × 10^−6^
*Lair1*	leukocyte-associated Ig-like receptor 1	−0.45	7.76 × 10^−6^
*Ccr5*	chemokine (C-C motif) receptor 5	−0.51	4.65 × 10^−6^
*Csf1r*	colony stimulating factor 1 receptor	−0.44	1.82 × 10^−5^
*Laptm5*	lysosomal-associated protein transmembrane 5	−0.39	2.32 × 10^−5^
*Pik3cg*	phosphatidylinositol-4,5-bisphosphate 3-kinase catalytic subunit gamma	−0.47	9.25 × 10^−5^
*Ddah1*	dimethylarginine dimethylaminohydrolase 1	−0.39	3.98 × 10^−5^
*Lcp2*	lymphocyte cytosolic protein 2	−0.36	4.94 × 10^−4^
*Arap2*	ArfGAP with RhoGAP domain, ankyrin repeat and PH domain 2	−0.39	4.40 × 10^−4^
*Lyz2*	lysozyme 2	−0.55	9.26 × 10^−4^
*Ctsd*	cathepsin D	−0.52	3.45 × 10^−4^
*Tlr7*	toll-like receptor 7	−0.51	1.19 × 10^−3^
*Inpp5d*	inositol polyphosphate-5-phosphatase D	−0.37	1.04 × 10^−3^
*Samsn1*	SAM domain, SH3 domain and nuclear localization signals, 1	−0.46	5.13 × 10^−3^
*Fyb*	FYN binding protein	−0.47	1.63 × 10^−3^
*Lrrc3*	leucine rich repeat containing 3	−0.38	1.43 × 10^−3^
*C3ar1*	complement component 3a receptor 1	−0.41	2.80 × 10^−3^
*Cd300a*	CD300A molecule	−0.50	4.39 × 10^−3^
*Plin2*	perilipin 2	−0.42	5.01 × 10^−4^
*Sorbs1*	sorbin and SH3 domain containing 1	−0.48	3.32 × 10^−3^
*Itgb3*	integrin beta 3	−0.36	1.46 × 10^−2^
*Slc5a7*	solute carrier family 5 (choline transporter), member 7	−0.69	1.81 × 10^−2^
*Tbc1d10c*	TBC1 domain family, member 10c	0.61	1.34 × 10^−2^
*Cbx2*	chromobox 2	0.54	6.52 × 10^−3^
*Ifitm1*	interferon induced transmembrane protein 1	0.60	6.54 × 10^−3^
*Cxcr5*	chemokine (C-X-C motif) receptor 5	0.75	6.55 × 10^−3^
*Pcgf2*	polycomb group ring finger 2	0.44	8.38 × 10^−5^

## Data Availability

Authors declare availability of data and material upon request.

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
