# Peer review of "The Leukotriene Receptor Antagonist Montelukast Attenuates Neuroinflammation and Affects Cognition in Transgenic 5xFAD Mice"

_ijms, 2021, doi:10.3390/ijms22052782_

Round 1

Reviewer 1 Report

On page 5, line 160, please provide the gene name as well as its role for the top three. Secondly with the ‘gene encoding for one of the CysLT receptors’ is mentioned but nowhere is it mentioned which receptor it encodes for. This might be important as on Fig 1 it is displayed that MTK only blocks CPR17 and CYSLT1R and not CysLT2R.

Page 19 figure 8, I appreciate the graphical summary but why is MTK not represented on the figure?

Grammatical and formatting changes:

You shouldn’t start a sentence with a number (page 5, line 157)

Tables 2-4 are spaced in a way that makes the tables harder to follow than they should be. Please increase column width to reduce orphan lines or provide horizontal lines to make it easier on the reader.

Page 8, line 213, this is a run-on sentence, please correct.

All figures need either titles on the figure or a clear title in the caption

There are a lot of different tests within this experiment, I would recommend utilizing more headings within your paper to help guide your reader through your experiments.

Author Response

  • On page 5, line 160, please provide the gene name as well as its role for the top three. Secondly with the ‘gene encoding for one of the CysLT receptors’ is mentioned but nowhere is it mentioned which receptor it encodes for. This might be important as on Fig 1 it is displayed that MTK only blocks CPR17 and CYSLT1R and not CysLT2R.

We thank reviewer 1 for this note and clarified that it is the receptor GPR17.

  • Page 19 figure 8, I appreciate the graphical summary but why is MTK not represented on the figure?

We agree with the reviewer and modified figure 8 to that end.

  • You shouldn’t start a sentence with a number (page 5, line 157)

We thank the reviewer for this input and rewrote the sentence, starting now with “In total, 59% .. etc”

  • Tables 2-4 are spaced in a way that makes the tables harder to follow than they should be. Please increase column width to reduce orphan lines or provide horizontal lines to make it easier on the reader.

We thank the reviewer for this comment. We added horizontal lines and increased column widths to make the tables clearer.

  • Page 8, line 213, this is a run-on sentence, please correct.

We totally agree and split this sentence into three shorter ones in the revised version of the manuscript.

  • All figures need either titles on the figure or a clear title in the caption

We agree with the reviewer and highlighted the titles of the figures in the captions

  • There are a lot of different tests within this experiment, I would recommend utilizing more headings within your paper to help guide your reader through your experiments.

We agree with the reviewer and added subheadings in the results section to supply more guidance for the reader.

Reviewer 2 Report

The manuscript ‘The leukotriene receptor antagonist Montelukast ameliorates neuroinflammation and improves cognition in a mouse model of Alzheimer’s disease’ investigates the effect of MTK on total gene expression (RNAseq), microglial and lymphocyte recruitment/activation and behaviour in the 5xFAD mouse model of AD.

While the activity of MKT is perhaps not novel given the cited references supporting its effect in neurodegeneration models (aged brain, dementia with lewy bodies) and retrospective patient studies, the novelty of this paper appears to reside in the delivery of this drug in the form of a buccal film in two different doses to 5xFAD mice at an already progressed state of pathology (advanced plaque pathology, cognitive deficits, impaired neurogenesis and progressed neuroinflammation).  The focus and experimental design of this paper was somewhat at odds with this aim given that no direct comparison was made between this form of drug delivery and conventional drug delivery; the study did not include wildtype/litter mate controls and there was no evidence provided beyond a reference to a bioRxiv manuscript for improved bioavailability and blood-brain-barrier penetrance for drug delivered by this method. 

Treatment began at 5 months, where plaque deposition is extensive and according to the literature there is evidence for impaired cognition.  Mice were treated with two doses of drug for approximately 10 weeks before cognitive testing and euthanised at 8 months for RNAseq analysis and pathology.

RNAseq analysis shows significant changes in gene expression however it must be noted that this study only included treated and untreated 5xFAD mice and so changes to gene expression can be attributed to the effects of MTK, but cannot be directly attributed to disease modification because the study did not include treatment of control mice.

The authors report sex related differences in gene expression. Female mice are known to be more severely affected in this 5xFAD model.  It is difficult to assess the significance of these differences without the gene expression profile of control mice.

Not unexpectedly, MTK treatment was associated with differential expression of genes relating to glial cell activation and neuroinflammation, given that it is an antagonist of the leukotriene receptor.  Again, these results would have been more informative if wildtype or litter mate controls had been used in the analysis. Non-the-less this was used as the basis for further investigation of microglial cells.

Relative to wildtype controls, 5xFAD have been reported to develop an age-related increase in microglia (F4/80) and Astrocytes (GFAP) that correlates with the development of amyloid plaques.

In the current study there was a small but significant increase in Iba1-IR cells that appeared to reflect an increased proportion of resident microglia (Tmem19+). I would suggest the term resident microglia rather than homeostatic microglial for this population (as used by the authors, which reflects an activation state which does not seem to be the case for this marker; Bennet et al).  It is not clear however, why treatment with MKT increases the total number of myeloid cells (pro-inflammatory) and specifically the resident microglial population.

Morphological analysis of soma size was used as a marker of microglial activation.  From this analysis MKT treatment significantly decreased the soma size of Iba+/Tmem19+ (resident microglial) but had no effect on the infiltrating macrophage population (Iba+/Tmem19-).  It is not clear why a comparison between the size of the different myeloid popultions has been performed as according to Bennett et al the Iba+/Tmem19+ and Iba+/Tmem19- population represents two different myeloid cell populations. It is interesting to compare the soma size of these two populations within the treatment or control groups.  But the comparison between untreated Iba+/Tmem19- and MKT treated Iba+/Tmem19+ does not seem appropriate.

Page 12, lines 307-309 and Figure 3 states that the percentage of 5-Lox+ cells in the Iba1+ microglia population was significantly decreased by approximately ten percent in the low and high dose treated groups (Fig 4F). However, the Iba+ represents both resident microglial and infiltrating macrophage populations. These statements should be clarified.

Page 12, paragraph commencing line 314.  The authors quantified plaque pathology and reported no differences in MTK treated versus untreated. In figure 3 the authors indicate that there is a lack of Tmem19 positive cells is the vicinity of the plaque in both vehicle and MTK treated mice.  Was there any difference in the distance of Tmem19- or Tmem19+ cells relative to the plaque, with treatment.

For behavioural studies although significant changes in learning and memory were reported for mice receiving MTK the overall significance of these effects cannot be determined without the inclusion of wildtype mice.  For example, in the MWM data presented in figure 7, the vehicle control 5xFAD mice do not appear to have developed impaired memory. 

To include behavioural data in this study naïve (and drug treated) control mice must be included to show impairment in 5xFAD mice and subsequent rescue by treatment with MTK.

Page 18, line 520 states ‘Our herein shown data further demonstrates that pharmacological inhibition of LT signalling reduces glial cell activity and leads to a reduced number of microglia that are involved in LT production.’  It is not clear how this statement relates to the data in figure 3 of a significant increase in numbers of Iba1+/Tmem119+ microglia with MTK and perhaps in relying in the data presented in figure for 5-lox expression in iba1+ cells (which as discussed above does not reflect only the microglial population).

The supplementary information was difficult to access and would benefit from figure legends and table titles.

It is not clear what information supp Table 2 conveys.

Supp Figure 2(f).  Authors should comment on the dose dependent increase in the Iba1+/Tmem19- cells in the cortex of female mice.

Supp Figure 3 (a) the authors should comment on the significantly reduced size of the Iba+/Tmem- population following MTK treatment in the cortex which was not observed in the hippocampus.

Round 2

Reviewer 2 Report

While I appreciate the authors contention that their ‘focus was on the effects of MTK treatment in two different doses compared to vehicle’ I do not agree with the argument that there approach is justified because it aligns with phase II clinical trials or the rationale that MTK had been administered in their previous study Marschallinger, Altendorfer et al. 2020 because the previous study did not include the RNASeq or characterisation of immune response, which they point out is unique to their study.  They further argue that     ‘we do not state in our manuscript that MTK restores the disease model to wildtype conditions in any kind, because we cannot proof this with our data’ yet statements such as

Line 30. The treatment increased the number of Tmem119+ microglia and down-regulated genes related to AD-associated microglia and to lipid  droplet accumulating microglia, suggesting that the MTK treatment targets and modulates microglia phenotypes.

Line 515 ‘However, the behavioral data from  our study shows that compared with placebo MTK treatment indeed attenuated behavioral deficits in female 5xFAD mice, which show cognitive deficits compared to their WT  littermates already at an age of 4-5 months 69.71. and indeed the title of the manuscript ‘The leukotriene receptor antagonist Montelukast ameliorates neuroinflammation and improves cognition in a mouse model Alzheimer’s disease’ does imply MTK restores the disease model to wildtype conditions’ and even if this is not the intention in is the impression.    

The authors may argue technically that by acknowledging that the control data is from other sources it is somewhat misleading and should be more implicitly stated. The reliance on previous publications for control data is also somewhat concerning given this study analysed 8-month-old mice and as a progressive disease difference might be expected based on age.

The manuscript needs to be carefully reviewed to ensure the data on which their observations are based are clearly described and the limitations ackoweldged.             

While a correction was made regarding statistical analysis in Figure 4, the text (lines 301-304) needs to be updated.

Author Response

Here, the point by point response to the reviewer. 

Reviewer: While I appreciate the authors contention that their ‘focus was on the effects of MTK treatment in two different doses compared to vehicle’ I do not agree with the argument that there approach is justified because it aligns with phase II clinical trials or the rationale that MTK had been administered in their previous study Marschallinger, Altendorfer et al. 2020 because the previous study did not include the RNASeq or characterisation of immune response, which they point out is unique to their study. They further argue that ‘we do not state in our manuscript that MTK restores the disease model to wildtype conditions in any kind, because we cannot proof this with our data’ yet statements such as 

Line 30. The treatment increased the number of Tmem119+ microglia and down-regulated genes related to AD-associated microglia and to lipid droplet accumulating microglia, suggesting that the MTK treatment targets and modulates microglia phenotypes.

Response: To clarify this sentence, we added (page1 line 25) that the effects were obtained in the disease model and compared to vehicle treatment.

Reviewer: Line 515 ‘However, the behavioral data from our study shows that compared with placebo MTK treatment indeed attenuated behavioral deficits in female 5xFAD mice, which show cognitive deficits compared to their WT littermates already at an age of 4-5 months 69.71.

Response: We agree that this sentence might be misunderstood in a way that it have been the exact same mice tested at 4-5 months of age prior to our experiment, which was not the case. The data on behavioral deficits at 4-5 months of age between WT and tg mice come from the original publication introducing the 5xFAD model (Oakley, Cole et al. 2006).

We altered the sentence into: “However, the behavioral data from our study shows that compared with placebo high doseMTK treatment indeed attenuated behavioral deficits in 8.5 months old female transgenic 5xFAD mice. In this model transgenic mice show cognitive deficits compared to their WT littermates already at an age of 4-5 months 71

Reviewer: and indeed the title of the manuscript ‘The leukotriene receptor antagonist Montelukast ameliorates neuroinflammation and improves cognition in a mouse model Alzheimer’s disease’ does imply MTK restores the disease model to wildtype conditions’ and even if this is not the intention in is the impression.    

Response: We do agree with the reviewer and thank the reviewer for the comment that the title could unintentionally be misleading. We changed the title to “The leukotriene receptor antagonist Montelukast attenuates neuroinflammation and affects cognition in transgenic 5xFAD mice. Furthermore, we additionally stated in the abstract, that we used the disease phenotype compared to vehicle to clarify the limitations in this study already in the abstract.

Reviewer: The authors may argue technically that by acknowledging that the control data is from other sources it is somewhat misleading and should be more implicitly stated. The reliance on previous publications for control data is also somewhat concerning given this study analysed 8-month-old mice and as a progressive disease difference might be expected based on age.

The manuscript needs to be carefully reviewed to ensure the data on which their observations are based are clearly described and the limitations acknowledged.         

Response: We overworked the total manuscript with regard to the reviewers comment. We want to make sure it is clear that we used tg mice treated with vehicle as control in our study and all our results refer to the comparison of MTK treatment in tg mice versus vehicle treatment in tg mice. Clearly, a limitation of our study is that we can not evaluate if the effects of MTK result in WT conditions in the treated tg mice, because we did not include WT mice.

Line 151: we added that the hippocampal tissue used in the RNA Seq was from high dose and vehicle treated transgenic mice

Line 198: we clarified that the comparison was done between MTK and vehicle

Line 256: we clarified that the microglia analysis was done in transgenic mice

Line 260: we clarified that the comparison was done between MTK and vehicle

Line 325: we clarified that the comparison was done between MTK and vehicle

Line 327: We clarified that our results come from transgenic mice

Line 361f: we clarified that the comparison was done between MTK and vehicle

Line 377: we clarified that the behavioral analysis was done only in transgenic mice

Line 386: we clarified that the behavioral analysis was done only in transgenic mice

Line 402: we clarified that the behavioral analysis was done only in transgenic mice

Line 448f: we clarified that the behavioral analysis was done only in transgenic mice

Line 484f: we clarified that in this in vivo study we compared MTK and vehicle in transgenic mice

Line 553: We clarified that our results refer to transgenic mice

Furthermore, we added a statement at the beginning of the discussion on the limitations of our study (page 17, line 491ff)

Reviewer: While a correction was made regarding statistical analysis in Figure 4, the text (lines 301-304) needs to be updated.

Response: We thank the reviewer for this comment and updated the text.

Oakley, H., S. L. Cole, S. Logan, E. Maus, P. Shao, J. Craft, A. Guillozet-Bongaarts, M. Ohno, J. Disterhoft, L. Van Eldik, R. Berry and R. Vassar (2006). "Intraneuronal beta-amyloid aggregates, neurodegeneration, and neuron loss in transgenic mice with five familial Alzheimer's disease mutations: potential factors in amyloid plaque formation." J Neurosci26(40): 10129-10140.